# Establishing Integrative Framework for Sustainable Reef Conservation in Karimunjawa National Park, Indonesia

Agung Dwi Sutrisno [1,2], Yun-Ju Chen [3], I. Wayan Koko Suryawan [1,4] and Chun-Hung Lee [1,5,*]

1 Department of Natural Resources and Environmental Studies, National Dong Hwa University, No. 1, Sec. 2, Da Hsueh Rd. Shoufeng, Hualien 97401, Taiwan
2 Institut Teknologi Nasional, Jl. Babarsari, Caturtunggal, Depok, Sleman, Yogyakarta 55281, Indonesia
3 Department of Applied Economics, National Chung Hsing University, 145 Xingda Rd., South Dist., Taichung City 402, Taiwan
4 Department of Environmental Engineering, Faculty of Infrastructure Planning, Universitas Pertamina, Jakarta 12220, Indonesia
5 Center for Interdisciplinary Research on Ecology and Sustainability, College of Environmental Studies and Oceanography, National Dong Hwa University, Hualien 97401, Taiwan
* Correspondence: chlee@gms.ndhu.edu.tw; Tel.: +886-3-8903343

**Abstract:** The Coral Triangle region is facing negative impacts due to unbalanced carrying capacity and inappropriate public behavior, leading to unsustainable reef tourism. As a result, there has been increased awareness and preference for sustainable reef conservation (SRC). This study evaluates the integrative perspective framework of tourists' heterogeneity preferences in SRC programs using a choice experiment conducted in Karimunjawa National Park (KNP), Indonesia. The study found that tourists preferred boat anchoring at the mooring buoy, a lower number of boats, smaller tourist groups with interpretation, added information boards, and environmental awareness education. Additionally, this research revealed that most tourists preferred the alternative SRC program and had a heterogeneous preference, which showed different features among each group. The scenario of an integrative program generated the highest value compared to the "recreational management" and "institutional control" scenarios. This evidence can assist policymakers in adapting policies for SRC programs and in potentially securing conservation funds associated with enhancing the institutional aspects of carrying capacity and marine environmental education for sustainable marine development.

**Keywords:** sustainable marine development; public environmental education; resource allocation; integrative program; willingness to pay

## 1. Introduction

Coral reefs are among the world's most important ecosystems, providing habitat and nursery grounds for numerous species of marine life and supporting the livelihoods of millions of people worldwide [1]. The Coral Triangle, which spans the waters of Indonesia, Malaysia, the Philippines, Papua New Guinea, the Solomon Islands, and Timor-Leste, is home to more than three-quarters of the world's reef-building coral species [2]. The area is also renowned for its rich fishery and marine biodiversity. However, unsustainable tourism, fishing, and the destruction of coral reefs, mangroves, and seagrasses are causing negative impacts on the ecosystem services they provide [3,4]. As part of the Coral Triangle, Indonesia has the largest archipelagic nation in the world, with over one-seventh of the world's coral reefs [5]. Karimunjawa National Park (KNP), located in the Java Sea, is one of Indonesia's primary tourist destinations and is recognized as vital for conserving marine biodiversity [6]. However, KNP faces numerous challenges, including unsustainable tourism practices, limited environmental awareness, and inappropriate behavior, which negatively impact its conservation effort [7].

Sustainable reef conservation (SRC) is crucial for conservation efforts [8,9] in KNP. Establishing an integrative framework for sustainable reef conservation in KNP can provide a comprehensive and holistic approach to address these challenges. SRC is critical for maintaining the ecosystem services coral reefs provide, which are vital to marine biodiversity. Coral reefs are sensitive ecosystems, and their health and survival depend on a complex balance of environmental factors [10]. Human activities such as unsustainable tourism, fishing practices, and the destruction of coral reefs, mangroves, and seagrasses are significant threats to the health of coral reef ecosystems [11]. Therefore, SRC efforts must consider a wide range of factors to ensure that the efforts effectively maintain the health and survival of coral reef ecosystems.

One of the critical factors that must be considered in SRC is boat anchoring placement. Boat anchoring is standard in tourist areas, where boats are used for activities such as snorkeling, diving, and island hopping [12,13]. However, the improper placement of boats can significantly harm coral reefs [14] and destroy the habitats of marine life. Therefore, it is essential to establish proper boat anchoring protocols to minimize the negative impacts of boat anchoring on coral reefs [15]. In addition, the number of boats is another critical factor that must be considered in SRC. The greater the number of boats in an area, the greater the potential for damage to the coral reef ecosystem [16]. As such, it is essential to establish a maximum number of boats that can be present in a particular area at any given time. This will help to ensure that the impact of boat traffic on the coral reef ecosystem is minimized.

Similarly, the number of tourists must also be considered in SRC. The greater the number of tourists, the greater the potential for damage to the coral reef ecosystem [17]. Therefore, it is essential to establish a maximum number of tourists that can be present in a particular area at any given time. This will help to ensure that the impact of tourism on the coral reef ecosystem is minimized. Information boards are also critical in SRC. These boards provide information to tourists about the coral reef ecosystem, including its importance, vulnerability, and the steps they can take to help protect it. In addition, information boards can raise awareness among tourists about the need to protect coral reefs and the steps they can take to minimize their impact on the ecosystem [18]. Environmental awareness education is another crucial factor that must be considered in SRC. This involves educating tourists about the importance of the coral reef ecosystem, its vulnerability, and the steps they can take to help protect it [19,20]. Environmental awareness education can help raise awareness among tourists about the need to protect coral reefs [21] and the steps they can take to minimize their impact on the ecosystem. Overall, SRC efforts must consider a wide range of factors, including boat anchoring placement, the number of boats, the number of tourists, information boards, and environmental awareness education. By integrating these factors, SRC efforts can be more effective in maintaining the health and survival of coral reef ecosystems in the long term.

This study aims to address the challenges facing the KNP by establishing an integrative framework for sustainable reef conservation that incorporates the various stakeholders' preferences and concerns. As a result, the study offers insights into sustainable marine ecotourism and management for marine protected areas in the Coral Triangle region. In addition, the framework can serve as a model for other marine protected areas (MPAs) facing similar challenges in managing their resources sustainably. The framework would incorporate integrated attributes related to SRC programs in the MPA management field, using a choice experiment (CE) approach. The CE approach seeks to estimate tourists' heterogeneous preferences across demographic and attitude segments for SRC programs and propose a creative thinking and evaluation framework for the Coral Triangle area.

By focusing on the KNP, a significant contributor to the Coral Triangle's marine biodiversity, this study can offer insights and best practices for other MPAs in the region facing similar challenges. The framework developed in this study can help policymakers and managers better understand tourists' heterogeneity preferences across demographic and attitudes segments for SRC programs and, therefore, design and implement more effective conservation policies and management strategies. Additionally, this study can

help bridge the gap between scientific research and practical implementation in marine conservation. So often, research is conducted in isolation from real-world issues and challenges MPAs, and their management authorities face. This study's CE approach aims to incorporate the preferences of tourists in the decision-making process for SRC program design and implementation. This participatory approach can help increase buy-in and support from various stakeholders, which is critical to the success of conservation efforts.

## 2. Literature Review

### 2.1. Carrying Capacity

Carrying capacity is defined as at a single site, the optimal number of tourists who can make use of a tourism resource at any given point, using a variety of "scientific" methods [22]. Ayllon defines the maximum population that can be supported by a certain level of resources over a given period or the number of available habitats divided by the area of the individual expected for a given life stage [17]. At the same time, Zacarias et al. defined it as the optimal number of people who should be allowed without endangering the aspects of the ecological, social, and cultural environment [16].

Carrying capacity is a popular tool in tourism and recreation planning and management [17]. The tourism manager can estimate the anticipated number of visitors, facilities, and other services by knowing the carrying capacity. If the population exceeds the environment's carrying capacity, it is almost certain that the environment will suffer damage [16]. Therefore, carrying capacity is an important factor in conservation programs [17]. Moreover, visitors' feedback will positively impact the planning and management of sustainable tourism if carrying capacity is known. The coral reefs of KNP, as an ecosystem used as tourism objects, also have a specific carrying capacity. Less controlled tourism activities on coral reefs have the potential to exceed their carrying capacity and will have an impact on damage [6]. Damage is caused by a significant number of visitors [23], resulting in a large number of boats and mooring buoys (then anchoring on the reef slope) [24]. Therefore, restrictions on its use as a tourist area must be regulated according to the existing carrying capacity [25].

### 2.2. Public Environmental Education

Conservation in MPAs cannot carry out by only MPAs officials but requires the involvement of all stakeholders, both central and local governments, MPAs managers, tourist agency companies, communities, and non-government organizations (NGOs), including tourists [26]. Therefore, building environmental awareness is needed in conservation programs [15]. Public awareness of the environment, including coral reefs, is strongly influenced by tourists' knowledge. Tourists with higher education have higher environmental awareness as well [10]. Tourists involved in environmental or conservation groups also have a high level of awareness [10,15].

Therefore, providing education to tourists about the importance of caring for the environment (coral reefs) is essential [10,27]. Environmental education can provide to tourists through training, short courses, and briefings, either directly or through the media [13,28]. Media can be information boards [15], booklets, stickers, billboards, short videos [13] about environmental awareness, and other media. Therefore, educational efforts and campaign media for environmental awareness, especially concerning coral reefs, are needed [10,13].

### 2.3. Applying Carrying Capacity and Environmental Education to SRC

We use the SRC environment required to develop sustainable tourism activities based on carrying capacity and environmental education assessments to combine planning and tourist destination strategies (Figure 1). Carrying capacity studies seek to balance the environment conservation used for activities with management for long-term growth. In other words, they can be viewed as a strategic aspect for preserving a site's beauty [29].

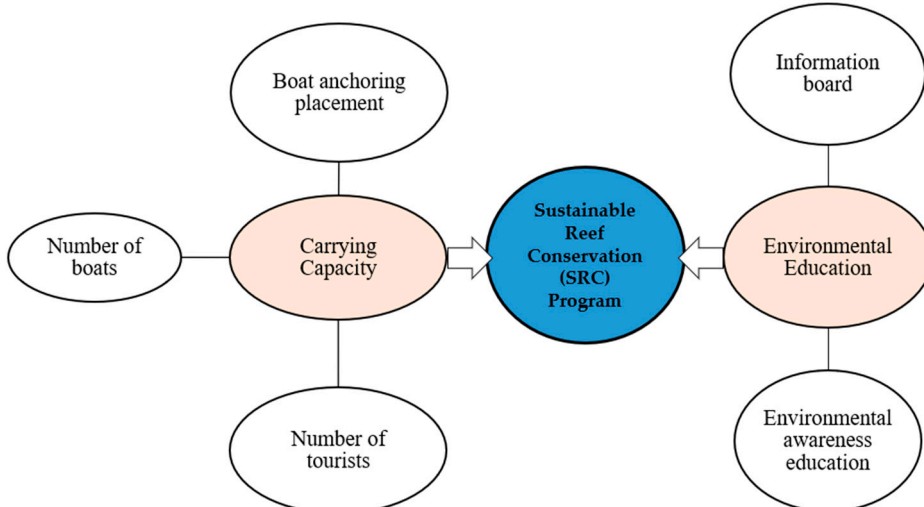

**Figure 1.** Applying carrying capacity and environmental education to SRC Program.

First, in carrying capacity, we use boat anchoring management. Boat anchoring management, defined as the short-term deployment of an anchor to the seabed to maintain a boat in one spot, can cause damage to the bottom, producing significant ecological repercussions [30,31]. When anchoring occurs in vulnerable environments such as coral reefs, possible anchor damage is mitigated by establishing no anchoring zones as part of the MPA [32]. Crowding is one issue that could explain why some boat anchor in sensitive areas [33]. Second, as the number of boats increases globally, there is a greater likelihood that mooring places will be fully occupied, resulting in a higher density of anchored yachts [34,35]. Finally, monitoring and managing the number of visitors could be achieved by developing supportability scenarios that consider ecosystem preservation, the rate of mangrove degradation, and the movement of megafauna and endemic species [36].

As a long-term approach, sustainable tourism that aims to maintain the landscape and provide and encourage environmental education might benefit protected areas [37,38]. First, an information board can provide a wealth of information essential for marine conservation and spatial planning [39]. This map identifies an optimal extension for the current MPA network by selecting the highest-ranked locations outside the current MPA network [39]. The second attribute may be regarded as educational in MPA for schools' educational tourism market. The bulk of empirical study and academic debate has concluded that educational tourism has the potential for hybridization with other segments of tourist and non-tourism sectors and to contribute to global peace [40].

## 3. Materials and Methods

### 3.1. Research Area

This research was conducted in the Karimunjawa National Park, Indonesia (Figure 2). This national park is a group of islands separate from the island of Java. This park has a total area of 111,625 ha, which includes 1507.7 ha of land area and a marine conservation area of 110,117.3 ha [5]. The KNP location can be reached by vessel or plane from Semarang (the capital of Central Java province), or it can also be reached by vessel from Jepara Sea Port. KNP is one of 54 national parks in Indonesia. KNP has been designated a national park by the Indonesian government since 1999 and is managed by the National Park Office. This park is used as a conservation area for the forest, mangroves, turtles, and coral reefs conservation area, as well as a tourist spot. Among the tourist areas are beaches, mangrove areas, turtle cultivation, religious tourism, and coral reef tourism [5].

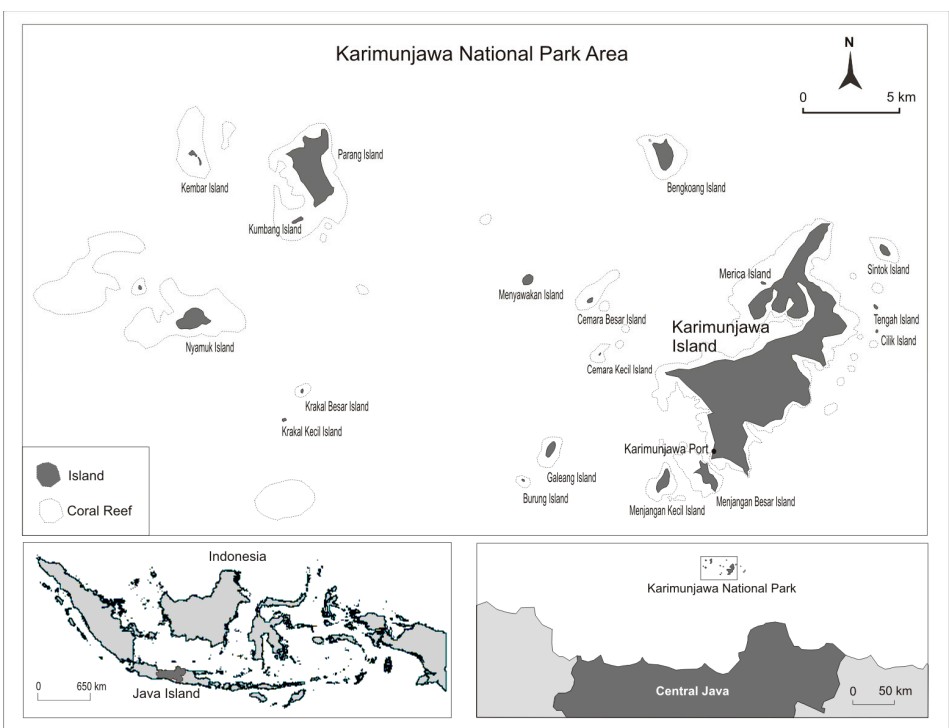

**Figure 2.** Map of Karimunjawa National Park, Jepara Regency, Indonesia.

Coral reef tourism is the most popular tourist destination in KNP. Of the 27 islands, 12 of them are designated for coral reef tourism [41]. However, over time this coral reef area has decreased in quality. If this is allowed, it will threaten the sustainability of coral reefs. That is why KNP is the main choice used as a research area.

### 3.2. The Attributes and Levels Design for SRC Program

To determine which SRC program a person preferred, we created a CE design based on the CE procedure [42,43]. First, we established the CE attributes based on the literature review of carrying capacity [10,44], public environmental education [10,13,15,28], and reef conservation program [10,15,20,21]. Then, we integrated the aspects of boat anchoring placement [26,45,46], number of boats [15,42,47], number of tourists [15,42,47], information boards [15,42,47], environmental awareness education [28,48], and conservation fund [10,12,15,20,21] (Table 1). Second, we discussed each of these attributes with various stakeholders, such as KNP managers, tourism associations, NGOs, and tour guides. These discussions and interviews are used to find out the current condition of each attribute and alternative expectations for management changes in the future. Lastly, we conducted a pretest to determine willingness to pay for conservation costs as a basis for designing a formal questionnaire. Based on the interviews and pre-test, we designed the attributes and levels as shown in Table 1.

Based on the attributes and levels in Table 1, the number of combinations was determined to be 540 ($2 \times 3 \times 3 \times 3 \times 2 \times 5$). We then used an orthogonal design to reduce unreasonable combinations [15,42] in the CE model study. The results of this orthogonal rationalization became alternative choices in the CE. The selected alternatives for the SRC program were randomly chosen from three options (a choice set), where every choice set had two alternatives and status quo options (Figure 3). Thus, we obtain a total of 40 versions of the choice set [12,42,43]. In addition to the choice-set questions, we also added two other sections. Part one is about traveling behavior and environmental awareness. Part two is about social background information.

**Table 1.** Attributes, levels, and variable names of KNP choice experiment design.

| Attributes | Levels | Variable Name |
|---|---|---|
| Boat Anchoring Placement | 1. Status quo: on the reef slope | ANCHOR |
| | 2. Put the boat anchoring on the mooring buoy | ANCHOR1 |
| Number of boats | 1. Status quo: no limited | BOAT |
| | 2. Set maximum 10 boats in a recreational activity | BOAT1 |
| | 3. Set maximum 5 boats in a recreational activity | BOAT2 |
| Number of tourists | 1. Status quo: per guide for 7 to10 tourists | TOUR |
| | 2. Per tour guide for 5 to7 tourists | TOUR1 |
| | 3. Per tour guide for 3 to5 tourists | TOUR2 |
| Information board | 1. Status quo: no information board | INFOR |
| | 2. Added more information boards in the KNP port | INFOR1 |
| | 3. Added more information boards in the KNP port and site | INFOR2 |
| Environmental awareness education | 1. Status quo: 5 min on the boat | EDU |
| | 2. Set more interaction in the classroom (at least 15 min) | EDU1 |
| Conservation fund [1] | 1. status quo: no conservation fee | Fund |
| | 2. increase to IDR 25,000 [2] | |
| | 3. increase to IDR 50,000 | |
| | 4. increase to IDR 75,000 | |
| | 5. increase to IDR 100,000 | |

[1] 1 USD eq 14.000 IDR. [2] The unit of conservation fund for the SRC program is per person per trip.

**Figure 3.** An example of choice experiment question.

### 3.3. CE Methodology

The CE method is a discrete choice model in which respondents are presented with a series of hypothetical scenarios, each with a different set of attributes, and asked to select their preferred option. [12,15,42]. The advantage of CE is that it can provide respondents with a variety of options from which to choose based on their preferences [42,47,49,50]. It can also analyze large amounts of data in a single application and can be used to estimate the effect of changes on various attributes [51]. So, we used the CE method to determine how different the tourists' preferences were in KNP, Indonesia, as part of the SRC program. In addition, the latent class model (LCM) is frequently used in choice experiments (CEs) to identify distinct groups of people with similar preferences for the attributes under consideration [15,52,53].

The 'tourists' preference function SRC can be shown with the following Equation:

$$V_{ij} = \beta_1 \text{Anchor}_j + \beta_2 \text{Boat}_j + \beta_3 \text{Tour}_j + \beta_4 \text{Infor}_j + \beta_5 \text{Edu}_j + \beta_6 \text{Fund}_j \quad (1)$$

where $V_{ij}$ is the preference function for SRC with the alternative $_j$ and all the other attributes and levels [7,43]. $\beta_1$ to $\beta_6$ are the estimated coefficients for alternative $_i$, where if a tourist chooses the status quo that identifies the value 1, or 0 otherwise [10,12]. The 'model's attributes are described as follows:

- $\text{Anchor}_j$: the attribute of boat anchoring placement, where Anchor = 1 means tourists would prefer to put the boat anchoring on the mooring buoy, otherwise Anchor = 0;
- $\text{Boat}_j$: the attribute about the number of boats in recreational activity in KNP, where Boat = 1 means that tourists prefer to set the carrying capacity of the number of the boat, otherwise Boat = 0;
- $\text{Tour}_j$: the attribute about the number of tourists in a recreational event, where Boat = 1 means that tourists prefer to set the carrying capacity of 'tourists' number, otherwise Tour = 0;
- $\text{Infor}_j$: this attribute is about the level of the information board, where Infor = 1 means that tourists prefer added more information boards in the KNP port and site, otherwise Infor = 0;
- $\text{Edu}_j$: this attribute is about the level of environmental awareness education, where Edu = 1 means that tourists prefer to choose more interaction about environmental education, otherwise Edu = 0;
- $\text{Fund}_j$: is the WTP for an SRC conservation fund (0, 25,000, 50,000, 75,000, and 100,000 Indonesian Rupiah). This WTP value is obtained based on the pre-test survey.

We can evaluate the marginal willingness to pay (MWTP) from the empirical results of the SRC program. The MWTP for the SRC program following the level change can be estimated by the ratios of the SRC attribute parameters and the conservation fund attribute and then written in Equation (2):

$$\text{MWTP} = \frac{-\beta_{attribute}}{\beta_{Fund}} \quad (2)$$

where $\beta_{\text{attribute}}$ is the 'attribute's coefficient with the 'tourists' preference for the SRC program, and $\beta_{\text{fund}}$ is the 'attribute's coefficient of the WTP [7,42,43]. Thus, we can evaluate the MWTP according to the two Equations.

### 3.4. Survey and Data

Under the criteria of 95% confidence interval and 5% estimation bias [49], all assume that tourists' SRC preference is equal for the KNP program; the sample size for our study was determined as 400 tourists. This research was conducted in July–November 2020. First, a pre-test was conducted on 50 respondents (local tourists) for the formal questionnaire design. Then, a formal survey was conducted on 400 local tourists who snorkeled in KNP. The local respondents were chosen because most visitors to the coral reef tourism in

KNP are local tourists [5]. The survey was conducted face-to-face at Karimunjawa Port, Semarang Port, and Jepara Port, on weekdays and during peak seasons.

According to the survey, male respondents outnumber female respondents (59%: 41%). Most tourists were married (50.2%), the majority were between 20 and 22 years old (55%), and most of the 'tourists' education was undergraduate (70%). They have a monthly income of 3–5 million IDR (214.3–357.1 USD) 51.2%, under 3 million IDR (214.3 USD) 44.5%, and above 5 million IDR (357.1 USD) 4.3%. The tourist occupation is dominated by students 28.5%, public employees 22%, teachers 14%, government employees 14%, freelancers 9.8%, business people 9.2%, and others 2.5% (See Table 2).

**Table 2.** Respondent demography of KNP tourists'.

| Characteristic | Frequency | % | Characteristic | Frequency | % |
|---|---|---|---|---|---|
| Male | 236 | 59 | Occupation | | |
| Female | 164 | 41 | Government employees | 56 | 14 |
| Single | 201 | 50.2 | Teacher/Lecture | 56 | 14 |
| Marriage | 199 | 49.8 | Freelance | 39 | 9.8 |
| Age | | | Public employees | 88 | 22 |
| <20 yrs | 8 | 2 | Bussiness | 37 | 9.2 |
| 20–29 yrs | 220 | 55 | Student | 114 | 28.5 |
| 30–39 yrs | 111 | 27.8 | Other | 10 | 2.5 |
| 40–49 yrs | 48 | 12 | Monthly Income Indonesian Rupiah (IDR) | | |
| >50 | 13 | 3.2 | <1 million | | |
| Education | | | 1–3 million | 25 | 6.3 |
| High school and under | 113 | 28.2 | 3–5 million | 153 | 38.2 |
| Bachelor | 280 | 70 | >5 million | 205 | 51.2 |
| Graduate and higher | 7 | 1.8 | | 17 | 4.3 |

## 4. Results and Discussion

### 4.1. Estimate Tourists' Preferences towards SRC in KNP

We estimate the group-specific results and heterogeneity preferences under latent class model (LCM) analysis. In this LCM analysis, we divided it into two classes to obtain more detailed information about the preferences and characteristics of different groups in the population. By understanding these differences in preferences and characteristics, policymakers can design policies that are more appropriate and tailored to the needs of each group, thereby improving the effectiveness and efficiency of the programs or policies implemented.

Based on Table 3, it can be seen that the results of the log-likelihood ratio (LLR) test have a greater value, which means that the SRC preference model is statistically at a significant level of 1% with better goodness of fit (GOF) [10,15,43].

The results of this study (Table 3) show that all respondents (classes 1 and 2) prefer anchor placement where it should be, namely the mooring buoy (ANCHOR1), reduction of the number of boats clustered at the site (BOAT2), and installation of information boards at ports and each site (INFOR1 and INFORM2). While many respondents (class 1, 62.2%) also prefer to reduce the ratio between tourists and guides (TOUR2), which is to be 3–5 tourists per 1 guide; in contrast, class 2 (minority, 37.8% of respondents) prefers the current conditions, with 7–10 tourists per guide. As for the preferences regarding environmental awareness education, all classes have the same preference: continue to educate on environmental awareness before recreation at the coral reef site.

**Table 3.** Result of LCM in KNP.

| Attributes and Levels | Class 1 (62.2%) | | MWTP (IDR) | Class 2 (37.8%) | | MWTP (IDR) |
|---|---|---|---|---|---|---|
| | Coefficient | Std. Error | | Coefficient | Std. Error | |
| ASC | −64.83 *** | 18.68 | | −1.35 *** | 0.52 | |
| ANCHOR | 19.90 *** | 5.81 | 2,532 | 0.33 ** | 0.13 | 27,050 |
| BOAT1 | −12.33 *** | 3.34 | | −0.2 | 0.13 | |
| BOAT2 | 12.33 *** | 2.69 | 208 | 0.26 * | 0.16 | 2030 |
| TOUR1 | −14.12 *** | 1.94 | | 0.05 | 0.15 | |
| TOUR2 | 8.26 *** | 1.51 | 353 | 0.14 | 0.17 | |
| INFOR1 | 0.87 | 4.4 | | 0.28 * | 0.16 | 43,915 |
| INFOR2 | 10.79 *** | 2.07 | 54 | 0.1 | 0.16 | |
| EDU | 3.62 | 3.14 | | 0.01 | 0.09 | |
| FEE | −1.72 *** | 0.23 | | −0.02 *** | 0 | |
| Class 1 Charactristic | | | | Model Properties | | |
| Constant | −1.75 *** | | | Log-likelihood function | −859.63 | |
| Monthly income > 3 million IDR | 1.56 ** | | | LLR | 917.41 | |
| Spend money 1–2 million IDR | 0.70 *** | | | Chi squared (0.01, 25] = 44.314 | | |
| First time to KNP | 0.94 * | | | | | |
| Thinking about KNP condition | −0.77 *** | | | | | |

Note: ***, **, * = > Significance at 1%, 5%, 10% level.

Respondents' preference for anchor placement in the mooring buoy (ANCHOR1) will positively impact KNP's carrying capacity because anchoring on reef slopes is clearly damaging to coral reefs [41]. By installing mooring buoy, the damage to coral reefs will be prevented [33]. The boats often make anchorages on the reef slopes because a mooring buoy is unavailable. For this reason, regulations are needed, so boats operating in coral reef areas are required to moor their boats at the mooring buoy [54–56]. This situation was also observed in the British Virgin Islands, where mooring buoy installation has impacted boats anchored on coral reefs [33]. Moreover, installing mooring buoys can also be used as community income by applying a mooring fee [21].

Limiting the number of boats at each site is very important, considering the number of boats will be directly proportional to the number of passengers, and the number of passengers will also be directly proportional to the intensity of contact with coral reefs [57]. Therefore, the respondent's preference for lowering the guides' ratio to passengers by 1:3–5 is very reasonable. Moreover, according to Akhmad et al. [42], one of the causes of reef damage in KNP is the intensity of snorkelers' contact with reefs. The smaller the number of snorkelers being guided, the more personalized the guidance can be. Increasing public awareness of SRC is crucial for increasing knowledge of coral reefs. One of the key findings of this study is the importance of providing briefings about coral reef awareness and installing information boards.. These findings support previous research highlighting the significance of awareness education [7,13,15,28]. This preference is also experienced by tourists in Oulanka National Park in Finland [42] and visitors to Khao Yai National Park in Thailand [15].

The model shows the 'respondents' monthly income [15,43], trip expenditure [10,47], trip experience [7,47], and perception of national park condition [6,17,58] affect their preference. Most tourists in class 1 have a high demographic income and spend money. Meanwhile, in terms of experience, they had litte, especially since they had only come to the KNP once and needed to think about efforts to improve the KNP. The interesting thing is that tourists who have experience tend to pay more.

Other research shows that tourist attraction is a driving factor that motivates tourists to visit a destination, especially because of the attractiveness of tourism products, required facilities, infrastructure, transportation, and hospitality hosting. Therefore, a total tourism product package that is expected, selected, perceived, and attracts tourists to make tourist visits is an important part of determining whether tourists will be satisfied or disappointed so that they can think of ways to improve tourist conditions.

This tends to be shown in respondents who are in class 2. Tourists in class 2 tend to have repeated visiting experiences, so they think about improving tourism conditions.

Where this is in class 2, tourists have a better cognitive response and a higher MWTP. Cognitive responses are perceptions and beliefs about an object, action, or condition that are compared with one's values, needs, wants, desires, and experiences [59]. Repeat visits are primarily due to a feeling of preference and a sense of interest in tourism activities [60], without anyone ordering, and these feelings become the basis for accepting the relationship between oneself and external objects (tourist objects), which is why repeat visits are difficult to refute, especially because the higher the level of liking, the higher the feeling of interest and the higher the understanding of tourist behavior, the greater the ability to form or produce "superior value" according to tourist needs. The higher the degree of conformity of the value offered with what tourists want, the higher the probability of success, the greater the probability of surviving in the market, and the market share will also increase.

### 4.2. Touristts' Heterogeneity in KNP and Their Preferences

A cross-tabulation analysis was also applied to determine the segmentation and heterogeneity of respondents, apart from the LCM analysis. The analysis results show that the items in the form of social background and respondent behavior, that are of significant value (<0.05), are marital status, income, the experience of visiting KNP, and snorkeling experience. In addition to this, the type of trip, environmental awareness education, and views of SRC conditions at KNP (first column, Table 4).

**Table 4.** Cross-Tabulation With Chi-Square Analysis based on delineated classes.

| Characteristic | All Respondents (400) | | Class 1 (62.2%) | | Class 2 (37.8) | | Chi-Square |
|---|---|---|---|---|---|---|---|
| | % | Num. | % | Num. | % | Num. | |
| **Marital Status** | | | | | | | |
| Married | 49.7 | 199 | 56.1 | 138 | 39.6 | 61 | 10.298 |
| Single | 50.3 | 201 | 43.9 | 108 | 60.4 | 93 | |
| Occupation | | | | | | | |
| Government employees | 14 | 56 | 21.6 | 45 | 5.3 | 11 | 26.766 |
| Teacher/lecture | 14 | 56 | 18.3 | 38 | 8.7 | 18 | |
| Freelance | 9.8 | 39 | 4.8 | 10 | 13.9 | 29 | |
| Public employees | 22 | 88 | 18.3 | 38 | 24.0 | 50 | |
| Bussiness | 9.2 | 37 | 6.3 | 13 | 11.5 | 24 | |
| Student | 28.5 | 114 | 29.3 | 61 | 25.5 | 53 | |
| Others | 2.5 | 10 | 1.4 | 3 | 3.4 | 7 | |
| Monthly income (IDR) | | | | | | | |
| <1 million | 6.3 | 25 | 2.0 | 5 | 13.0 | 20 | 51.752 |
| 1–3 million | 38.2 | 153 | 30.1 | 74 | 51.3 | 79 | |
| 3–5 million | 51.2 | 205 | 63.0 | 155 | 32.5 | 50 | |
| >5 million | 4.3 | 17 | 4.9 | 12 | 3.2 | 5 | |
| First time Visit KNP | 95.3 | 381 | 98.0 | 241 | 90.9 | 140 | 10.429 |
| More than one time Visit KNP | 4.8 | 19 | 2.0 | 5 | 9.1 | 14 | |
| Spent money | | | | | | | |
| <500 thousand | 3 | 12 | 0.0 | 0 | 7.8 | 12 | 92.945 |
| 500 thousand–1 million | 20.8 | 83 | 7.7 | 19 | 41.6 | 64 | |
| 1–2 million | 76.2 | 305 | 92.3 | 227 | 50.6 | 78 | |
| Trip organized by tourist | 7 | 28 | 3.7 | 9 | 12.3 | 19 | 10.959 |
| Trip organized by travel agency | 93 | 372 | 96.3 | 237 | 87.7 | 135 | |
| Have been snorkeling before | 8.5 | 34 | 3.7 | 9 | 16.2 | 25 | 19.257 |
| Have not been snorkeling before | 91.5 | 366 | 96.3 | 237 | 83.8 | 129 | |
| Get environmental education awareness before snorkeling | 95.8 | 383 | 99.6 | 245 | 89.6 | 138 | 23.195 |
| Not get environmental education awareness before snorkeling | 4.2 | 17 | 0.4 | 1 | 10.4 | 16 | |
| KNP needs to improve | 76.8 | 307 | 72.8 | 179 | 83.1 | 128 | 23.195 |
| KNP has no need to improve | 23.2 | 93 | 27.2 | 67 | 16.9 | 26 | |

Based on Table 4, respondents in class 1 are those who make their first visit (98%), are married (56.1%), the majority are workers (69.3%) and have set aside 1–2 million rupiah for their trip (71.4–142.8 USD). In addition, respondents also traveled through tourist agencies, had received education on environmental awareness, had never snorkeled, and thought that the KNP needed to improve the SRC. On the other hand, class 2 consists of respondents whose majority of income is lower than class 1. The majority are not married, but 71.1% of the respondents are equally dominated by those already working. Furthermore, it was the tourists' first visit to KNP, arranged through tourist agencies, during which they

received education on environmental awareness, went snorkeling, and concluded that there is room for improvement in KNP's SRC initiatives. LCM is very useful for analyzing market segmentation based on heterogeneity [10,15,42,49,54]. Based on this segmentation, the treatment of tourists by segment can be regulated by the KNP manager. For example, setting up sites for beginner and experienced snorkelers [10].

Based on Table 3 LCM above, it can also be seen that tourists who prefer to improve SRC conditions at the KNP are those who have a high income and expenditure and have never been to the KNP before. High-income tourists may place a higher value on reef sustainability because they have a better chance of obtaining an education. This education may familiarize them with conservation biology, environmental science, or sustainable tourism practices. So, they might better understand the bad things that happen when reefs get damaged and be more likely to help with conservation efforts. Moreover, tourists with more money may be more likely to do things that are good for the environment, like recycle, take public transportation, or buy products that are good for the environment. This may affect how they choose to travel, leading them to look for places and things to do that align with their values. As a result, they may be more likely to select a tour operator or hotel that places a premium on reef sustainability. High-income tourists may also be more likely to do things that are good for the environment because they can afford to. For example, they may be willing to pay more for a tour that uses electric boats rather than diesel-powered ones, or they may be willing to pay more to stay in a hotel with eco-friendly practices such as water conservation measures or renewable energy sources.

This preference for improvement in SRC in the high-income and high-expenditure tourist segments reinforces the previous findings. Juutinen et al. [42] concluded that high-income tourists preferred improved biodiversity and recreational services at Oulanka National Park in Finland. Sriarkarin and Lee [15] also found that high-income tourists in Khao Yai National Park in Thailand support reducing tourism effects and the development of tourism facilities in national parks. The findings of Lee et al. [10] regarding tourists in the reef recreation area in Kenting, Taiwan, also show that high-income tourists support changes in impact reduction. Likewise, high-income residents around the Danongdafu Forest Park in Taiwan support changes in the ecosystem service and land use programs [54]. Yin et al. [43] also found that residents and high-income tourists in Kinmen, Taiwan, supported an improved agricultural ecosystem function. This is also consistent with the findings of Yin et al. [7] that first-time and high-income tourists support quality improvements in the South Penghu Marine National Park, Taiwan.

Finally, park managers can use this heterogeneity finding to strategize park policies to achieve SRC in KNP. We provide a management framework based on these empirical results to evaluate SRP in KNP (Figure 4).

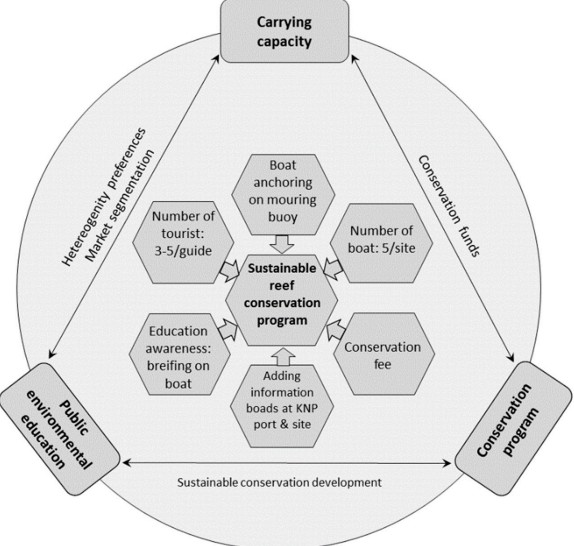

**Figure 4.** Conceptual framework on SRC programs.

### 4.3. Welfare Effect

Based on the LCM analysis presented in Table 3, it is evident that tourists prefer the SRC scenario which includes boat anchoring on the mooring buoy, set a maximum of 5 boats in a recreational activity, a limit of 3 to 5 tourists per tour guide, and additional information boards at the KNP port and site. This scenario is the most optimal among scenarios 1–3 outlined in Table 5. This means that, in an integrated sense, all attributes in the alternative changes proposed become tourists' preferences. If this scenario is implemented, the welfare effect of the change will reach a maximum conservation value of 76,142 ID (5.4 USD) per person per visit.

**Table 5.** Welfare effect for sustainable reef conservation in KNP.

| Attributes | Scenario 1 | Scenario 2 | Scenario 3 |
|---|---|---|---|
| | Recreational Management | Institutional Control | Integrative Programs |
| Boat Anchoring Placement | Put the boat anchoring on the mooring buoy | Put the boat anchoring on the mooring buoy | Put the boat anchoring on the mooring buoy |
| Number of boats | Set maximum 5 boats in a recreational activity | - | Set maximum 5 boats in a recreational activity |
| Number of tourists | Per tour guide for 5 to 7 tourists | - | Per tour guide for 3 to 5 tourists |
| Information board | - | Added more information boards in the KNP port | Added more information boards in the KNP port and site |
| MWTP | 208–32,173 IDR | 27,050–73,497 IDR | 54–76,142 IDR |

## 5. Conclusions

This study provides empirical results for an evaluation and management framework. Based on the above results and discussion, it can be concluded that tourists, in general, want changes in the sustainable coral reef conservation program at KNP. Their preferred changes include focusing on boat anchoring, restrictions on the number of boats, reducing the number of tourists per guide, and adding information boards for coral reef awareness. However, they felt that education on caring for coral reefs before snorkeling activities was sufficient a brief explanation on the boat. Regarding anchoring, they preferred that the boat be anchored at the mooring buoy. They suggested limiting the number of boats to five per site, with each guide responsible for three and five tourists. Tourists expected information boards on coral reef conservation to be installed at the KNP port because it is where they gather before spreading to various coral reef sites. They also preferred information boards to be added at each coral reef site.

People's heterogeneity has a significant impact on their preferences, as demonstrated in this research. Tourists who have high monthly incomes and spend more on trips show a preference for changes in sustainable reef conservation programs.. Similarly, those who have received information about coral reef conservation, joined conservation groups, and are visiting KNP for the first time also express a desire to change the sustainable coral reef conservation program. In addition to expressing a preference for changes to current practices, visitors are willing to pay more for measures such as changes to anchoring practices, limitations on the number of boats and tourists per guide, and the provision of an informational board.

Therefore, we suggest that managers can take the following steps. First, MPAs managers can integrate the boat anchoring placement, set control on the number of boats, set the number of tourists, and add the information board. They can also build up environmental awareness education and a conservation fund for MPA management as aspects of effective frameworks for sustainable marine management. This integration will increase the environmental carrying capacity of KNP. With the increase in environmental carrying capacity, the sustainability of coral reefs and MPAs will also increase, leading to greater conservation achievements. Second, MPA managers can create suitable target segmentation and resource allocation strategies under heterogeneous preferences and demographics

for the SRC program, combining the qualitative and quantitative data under systematic frameworks. This target segmentation is solely to realize SRC because each tourist segment has different habits, responses, and awareness toward conservation. Third, MPA managers can focus on the tourists who are married, have higher income, are first-time visitors, have higher travel expenditures, and have higher environmental education awareness. These groups had the highest preference and MWTP for the SRC attributes, and focusing on them could promote conservation efforts in the SRC program. Focusing on these groups is important, considering that SRC is the goal, and this group has the potential to support funds in SRC efforts. Finally, for the targeting and segmentation of an SRC program for sustainable marine development, the managers may focus on boat anchoring at the mooring buoy, limit the number of boats, have smaller tourist group with interpretation, add information boards, provide environmental awareness education before recreational activities, and establish an SRC conservation fund with the goals of sustainable marine development simultaneously in MPAs.

The findings of this study on the monetary value of protecting KNP's coral reefs have important implications for the development of environmental policies in the contect of SRC. These values can serve as a valuable baseline to monitor changes in response to poli-interventions such as restrictions on anchoring, limiting the number of boats and visitors per guide, and adding an informational boards. Such findings can help policymakers analyze the impact of human activities on coral reefs and make more informed decisions. Additionally, there is a need for a comprehensive approach to managing SRC in KNP, including implementing a carrying capacity limit and an environmental education campaign to increase awareness of the importance of coral reefs to the ecosystem and environmental sustainability. However, additional research is required to supplement the material and results. Future research could investigate the economic effects of additional activities such as snorkeling and recreational use of the park's islands' beaches. Furthermore, studies on habitat and fish protection, as well as community adaptation to the impact of environmental change on tourism, should be conducted. Finally, obtaining complete information on economic values is critical to implementing educated policy measures to conserve coral ecosystems in MPAs.

**Author Contributions:** Conceptualization, A.D.S. and C.-H.L.; formal analysis, A.D.S. and C.-H.L.; data curation, A.D.S.; investigation, A.D.S.; visualisation, A.D.S. and I.W.K.S., methodology, A.D.S., C.-H.L. and Y.-J.C.; Resources, A.D.S.; writing—original draft preparation, A.D.S., I.W.K.S., C.-H.L. and Y.-J.C.; writing—review & editing, C.-H.L. and Y.-J.C. All authors have read and agreed to the published version of the manuscript.

**Funding:** This work was supported by the National Science and Technology Council (NSTC), Taiwan [NSTC 109-2628-M-259-001-MY3].

**Institutional Review Board Statement:** Not applicable. However, the research group obtained informed consent from all participants and assured them that there would be no harm from either participating or not participating in the study. The participants' information is anonymous, confidential, and will not be disclosed.

**Informed Consent Statement:** Informed consent was obtained from all subjects involved in the study.

**Data Availability Statement:** The source of illustration in SRC is from the website of MDPI.

**Acknowledgments:** The authors would like to thank the Manager and staff of Karimunjawa National Park, the Head of the Wildlife Conservation Society (WCS) Indonesia-Karimunjawa, and several local tourism agency owners who cannot be mentioned individually for their cooperation during the research.

**Conflicts of Interest:** The authors declare no conflict of interest.

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
