# Peer review of "Establishing Integrative Framework for Sustainable Reef Conservation in Karimunjawa National Park, Indonesia"

_water, doi:10.3390/w15091784_

Round 1

Reviewer 1 Report (Previous Reviewer 1)

The manuscript has improved.

Author Response

Thanks for reviewers' comments and engagements.

Reviewer 2 Report (New Reviewer)

The submitted paper entitled “Establishing integrative framework for Sustainable Reef Conservation in Karimunjawa National Park. Indonesia” is an interesting paper and in my opinion can contribute to the international conversation regarding the management of marine protected areas. It uses an interesting analysis that could be implemented in other corresponding cases all over the world.

Nevertheless, there are some -general and specific- points in the text that need to be reconsidered and/or corrected in order for the paper to proceed for publication. Therefore, the authors need to take into consideration the following suggestions:

General considerations

·    Although the Introduction, Methodology and Conclusion sections are rather well written, the information presented in the Results – Discussion section often needs to be clarified as it appears rather blurry and misguiding. The fact that the authors try to combine in the same section both results and discussion leads to this result. The authors are urgently advised to rephrase and clear most of the content in this section.

·    In the Methodology section, the information in sub-sections 3.2.1 to 3.2.6 should be presented in just one 4-5 sentence paragraph without the use of references. The “extensive” information given here with the use of references could be transferred to the Discussion. 

·    Sub-section 4.3 (Welfare effect) seems to be more like Methodology than Results. Which part of the paragraph is exactly the results? If there are any, please clarify them. 

·    Table 5 seems to be more like a table providing methodological information than results. If this is the case, I would urge the authors to reconsider using it in the Methodology section. 

·    Figure 4 should move from the Conclusions to the Results section.

·    The word “Number” is often met with a capital letter (N) in several sentences (i.e., lines 112, 115, 117, 120, 161, 163, 205, 220, 221. 223, 226, 279, 282, 333, 350, 351, 356, 372, 476, 479, 492, 496, 521, 522)

Specific considerations:

Introduction

·    Lines 80-82: repetition of the info and therefore should be omitted. 

·    Line 90: move from line 92 the full name of MPAs

·    Lines 128-130: rephrase sentence

·    Figure 1: why two separate references to Environmental Education & Environmental Awareness Education? Don’t they have the same meaning?

Materials & Methods

·    Line 182: …land area and marine…: omit space

·    Lines 187-189: rephrase as following: “The park is used as a tourist destination, as a conservation…area as well as a tourist spot.”

·    Lines 192-196. The way this sentence is written, should not be in the Methodology section. Rephrase it to combine the relevant info with the last sentence.

·    In 3.2 use past tense in verbs “create”, establish”, “integrate”, “are”.

·    Line 202: delete “the CE procedure” as repetitive. 

·    Lines 211-212: see general consideration for creating one last paragraph with the corresponding info from 3.2.1 to 3.2.6.

·    Lines 252-253: clarify sentence.

·    Line 257: delete second full stop.

·    Line 271: delete full stop before parenthesis

·    Line 290: add explanation for WTP

·    Line 302: “was conducted”

·    Line 304: “was conducted on 400…”

·    Line 304: “snorkeled”

·    Line 308: repetition of sentence

·    Lines 309-313: rephrase sentence. Report only high values as the details are shown in Table 2. 

·    Line 313: convert national currency to US dollars. 

Results and Discussions

·    Line 318: Results and Discussion

·    Line 320: insert “Latent Class Model before (LCM)”

·    Line 329: delete “which”

·    Line 384: “needs, wants, desires, and experiences”

·    Line 400: delete semi-colon

·    Lines 403-404: which percentage goes where? Clarify sentence.

·    Line 419: “…because they have a better…”

·    Line 423: add space between the two sentences

·    Line 428: add space between the two sentences

·    Line 445: “first-time visits and high-income earners” a verb is missing

·    Table 4: there is no mention of the Table in the text

·    Line 451: “We provided three…”

·    Line 452: delete “of” scenarios

Conclusions

·    avoid the use of values in the Conclusions section (see line 480)

·    avoid the use of literature in the Conclusions section (see lines 498, 504, 513)

·    Line 486: put a semi-colon after “research”

·    avoid currency values in the Conclusions section (see line 493) and rephrase sentence

·    Line 496: Start sentence with a capital letter

·    Lines 518-528: repetition of information

Minor editing of English language required

Author Response

Dear Reviewer, we revised the manuscript base on the comments and suggestions , many thanks for your assistant and comments.

This manuscript is a resubmission of an earlier submission. The following is a list of the peer review reports and author responses from that submission.

Round 1

Reviewer 1 Report

This study focuses on health and sustainability of coral reefs in Karimunjawa National Park, which provide income for people in six countries. The work follows a methodology where an environmental trust funds is used as a financial attribute to assess the WTP of respondents for natural resources. The authors select a pre-test sample of 50 and sample size of 400 local tourists. Analyses of the regression results regarding choices is followed by policy recommendations to improve sustainability. While this is an important topic and the empirical work is fairly well executed, the paper is poorly written and is difficult to follow because of some long and poorly structured sentences. I understand that English may not be the primary language of the authors. In that case, they could consider seeking editorial assistance. Here are some examples of sloppy writing:

1. Headings are mixed up; Section 3.2 in line 177, then section number is repeated in line 235.

2. Grammar and spelling are a major concern throughout the paper. For example,

Lines 45-51, contain a long confusing sentence.

Lines 51-53, the sentence appears to be incomplete or poorly worded. Also, I suppose that by ‘consentrate’ in line 52, the authors mean ‘concentrate.’

Lines 62-66, is another long and confusing sentence. Also, ‘current’ is a better word in this context than ‘present.’

Lines 239-241, the sentence does not make sense.

Equation (1), I wonder if Vj should be replaced with Vij.

Line 410, I suspect ‘too’ is meant to be ‘two’.

I also have a couple of comments on the content of the paper as following.

1. Elaborate on why high-income (high-spending) tourists value sustainability of reefs more than the low-income (low-spending) tourists. Is this reflecting purchasing power or it captures education and environmental awareness?

2. What is the importance of class 1 and 2 in the empirical work for policy purposes?

Reviewer 2 Report

The paper is hard to follow. English problem obviously, but also content. It looks like a very simple study though, but burried in the Introduction under too much vague concepts that are therefore used with poor articulation to what is done. It will be much more easy to follow if the paper was just, in Introduction, focusing on the survey done and its goal, withtout all the overarching SRC, ES, CE, that are poorly connected and presented. line 72-74 is a good example of the sentences that do not help.

A litterature review is said used (line 62), which I assume is presented chapter 2. But this review does not follow some simple Reviews guidelines: how many papers used, which keywords were used to search for references, for which bibliographic databases...etc.

I actually give up trying to comment more in detail the paper as Methods are also so unclear. The choice of the various criteria are poorly argued.

Strangely, the resuts-Discussion are much more clearer, and in almost perfect English. It looks like a different paper. Hence I suggest that however wrote this last section also revises the Intro and Material-Methods sections.